# Relationship of Anthropometric Indicators of General and Abdominal Obesity with Hypertension and Their Predictive Performance among Albanians: A Nationwide Cross-Sectional Study

**DOI:** 10.3390/nu13103373

**Published:** 2021-09-25

**Authors:** Mohammad Redwanul Islam, Md Moinuddin, Samaha Masroor Saqib, Syed Moshfiqur Rahman

**Affiliations:** 1Department of Women’s and Children’s Health, Uppsala University, 75237 Uppsala, Sweden; mohammadredwanul.islam@kbh.uu.se (M.R.I.); samaha_saqib@yahoo.com (S.M.S.); 2Faculty of Health, Social Care & Medicine, Edge Hill University, Ormskirk L39 4QP, UK; mohammed.moinuddin@edgehill.ac.uk; 3Institute of Child Health, University College London, London WC1N 1EH, UK

**Keywords:** hypertension, obesity, anthropometric indicators, ROC curve analysis, optimal cut-off

## Abstract

Anthropometric indicators of general and abdominal obesity can predict cardiovascular disease outcomes. Their performance in predicting hypertension (HTN) varies across populations. We aimed to analyze the relationship of body mass index (BMI), waist circumference (WC), waist-to-height ratio (WHtR) and conicity index (CI) with HTN, to examine their predictive performance and to determine their optimal cut-offs in a nationally representative sample of Albanians aged 15–59 years (*n* = 20,635). Logistic regression models were fitted and sex-specific receiver-operating characteristic (ROC) curves were constructed. The indicators were positively associated with HTN. Sex modified the relationships, as associations appeared significantly stronger among females than males in the highest categories of the indicators. The area under ROC curves (AUCs) for BMI were 0.729 (95% confidence interval (CI): 0.720–0.738) among females and 0.648 (95% CI: 0.633–0.663) among males, and AUCs for WHtR were 0.725 (95% CI: 0.716–0.734) among females and 0.637 (95% CI: 0.622–0.652) among males. However, the AUCs for BMI and WHtR did not differ significantly among females (*p* = 0.279) and males (*p* = 0.227). BMI outperformed WC and CI in both sexes. The optimal BMI cut-offs were 27.0 kg/m^2^ among females and 25.6 kg/m^2^ among males, and that for WHtR were 0.53 among females and 0.54 among males. BMI and WHtR demonstrated similar discriminatory power, and the identified cut-offs may inform initiatives for structured HTN screening in Albania.

## 1. Introduction

Elevated blood pressure (BP) continues to be a global health concern. According to the Global Burden of Diseases, Injuries, and Risk Factors Study (GBD) 2019, elevated systolic blood pressure (SBP) tops a list of 87 risk factors accounting for an estimated 10.8 million deaths [1]. The number of adults with hypertension is predicted to reach 1.56 billion by 2025 [2]. However, the global burden has gradually concentrated in low- and middle-income countries over the last couple of decades [3]. On the other hand, five million estimated deaths are attributed to body mass index (BMI) greater than 20–25 kg/m^2^ [1]. In terms of risk-weighted prevalence, the global exposure to high BMI increased by 70.4% between 1990 and 2017, the highest among all risk factors evaluated in GBD 2017 [4]. This is particularly problematic for transitioning, middle-income countries as they experience obesogenic changes in dietary and physical activity patterns at the population level concurrent with socio-economic development. Unsurprisingly, elevated BP is the first, and high BMI the third leading health risk in Albania, an upper-middle-income country in southern Europe [5]. Linear dose-response meta-analysis documents the risk of hypertension (HTN) to escalate by 49% per five-unit increase in BMI [6]. The convergence of overweight/obesity and HTN necessitates examination of the performance of common anthropometric indicators of general and abdominal adiposity in tracking those with heightened cardiovascular risk.

BMI is extensively used for defining overweight and obese in research and practice. Nevertheless, BMI does not capture regional fat distribution and body proportions, and their variations across populations [7,8]. While central adiposity shows considerably stronger association with cardiometabolic risk than general obesity [9], BMI cannot gauge the former. Waist-to-height ratio (WHtR) addresses this limitation to a certain extent by correcting the WC for the height of individuals. Additionally, conicity index (CI) has been proposed [10] based on the assumption that higher central adiposity makes morphological profile of human body biconic (i.e., like a double cone with a common base) [11]. The higher the CI, the more centralized is the fat distribution. Although some studies suggest CI as a good indicator of cardiovascular risk [11,12,13], lack of established cut-offs limits its widespread application.

The above-mentioned indicators offer simplicity, noninvasiveness and potential for integration into primary care, but the magnitude of their associations with HTN, and their discriminatory capability, vary across populations [6,9]. To the best of our knowledge, HTN screening performance of these indicators has not been examined in a nationally representative sample from Albania. This is a critical research gap considering the burden of HTN, suboptimal BP control and lack of awareness in Albania [14]. Our objectives were three-fold: to explore the relationship of BMI, WHtR, WC and CI with HTN among Albanians, to examine the performance of these indicators in predicting HTN from analysis of receiver-operating characteristic (ROC) curves, and to determine their optimal cut-offs for predicting the likelihood of HTN.

## 2. Materials and Methods

### 2.1. Study Population

This cross-sectional study was based on nationally representative data from the 2017-18 Albania Demographic and Health Survey (ADHS). The National Institute of Statistics (INSTAT) and the Institute of Public Health (IPH) carried out this nationwide, household survey from September 2017 to February 2018 with technical assistance from ICF through the DHS Program [15]. The ADHS adopted a two-stage, stratified sampling strategy to ensure representativeness at national and subnational levels. A total of 16,955 households were selected, and 15,823 (93.3%) were successfully surveyed. Women aged 15–59 years in all surveyed households and men aged 15–59 in a subsample of 50% of those households underwent BP measurement and anthropometric assessment. Appendix A demonstrates the flow of participants into current study. Data on socio-demographic attributes, chronic disease status and lifestyle were also collected using household, women’s, and men’s questionnaires [15].

### 2.2. BP Measurement and Definition of Hypertension

Trained interviewers measured sitting BP with a manual sphygmomanometer and stethoscope. Three measurements were taken: one at the beginning, one in the middle and one at the end of the household survey. The average of the second and third measurements was used for defining HTN. Participants with SBP ≥ 140 and/or diastolic blood pressure (DBP) ≥ 90 mmHg were considered hypertensives according to the 2018 European Society of Cardiology (ESC)/European Society of Hypertension (ESH) Guidelines [16]. Additionally, participants were considered hypertensive regardless of their BP if they were on antihypertensive(s) or undergoing lifestyle modification for managing HTN.

### 2.3. Anthropometric Assessment and Calculation of Indicators

Height was measured with a portable ShorrBoard to the nearest 0.1 cm. Weight was measured to the nearest 0.1 kg using a Seca 878 scale, with participants barefoot and wearing light indoor clothes (e.g., t-shirt, trousers). BMI was calculated by dividing the weight in kilograms with the square of the height in meters. Participants were categorized into underweight (BMI < 18.5 kg/m^2^), normal-weight (BMI = 18.5–24 kg/m^2^), overweight (BMI 25–29 kg/m^2^), and obese (BMI ≥ 30 kg/m^2^) [17]. WC was measured with anthropometric tape placed horizontally, midway between costal margin and iliac crest, with participants standing. The reading was obtained at the end of gentle expiration. WHtR was the ratio of WC (cm) to Height (cm). CI was calculated from the Valdez equation: WC(m)/[0.109 × √{weight (kg)/height(m)}] [10].

### 2.4. Assessment of Covariates

Socio-demographic factors included in the analysis were age, sex, educational and socioeconomic status, and place of residence. Five age groups (15–19, 20–29, 30–39, 40–49, and 50–59 years) were created. Wealth index was derived from principal component analysis of data on household ownership of context-specific durable assets and such dwelling characteristics such as source of drinking water, toilet facility, and flooring material used. This captured socioeconomic status (SES) at the household level. Wealth indices were divided into tertiles: the highest tertile representing the richest, intermediate tertile the middle-status, and the lowest tertile the poorest households in Albania. Educational status was categorized into primary or below, secondary, and higher based on the highest level of education attended. Place of residence had two categories: urban (areas with at least 5000 residents) and rural. Self-reported smoking status was ascertained by a question asking if the participants had been smoking cigarettes or tobacco in the form of cigars, pipes, cheroots or cigarillos during the survey years. Alcohol consumption was self-reported based on responses to a question asking if the participants had consumed beer, wine, raki or other spirits in the preceding 12 months. History of diabetes was a self-reported binary variable (yes/no) as well.

### 2.5. Statistical Analysis

Statistical analysis was performed using Stata version 14.0 (StataCorp, College Station, TX, USA). We used the “svyset” command that allows application of sampling weight and accommodates household as the primary sampling unit, and thus accounts for the complex survey design. As males underwent BP and anthropometric assessment in 50% of the surveyed households, there was an imbalance between the number of females and males in the sample. To maintain consistency of analysis, we recalculated the sampling weights based on the household male-female distribution as a reference because it represents the male-female distribution in the general population. This newly constructed weight was combined with the sampling weight provided in the dataset for making the sample representative at national and sub-national levels.

Distributions of continuous variables were checked by visual examination of histograms and quantile-quantile plots, and were considered approximately normal. We calculated mean with standard deviation (SD) for continuous variables and frequency with percentage for categorical variables. We constructed sex-specific scatterplots of SBP and DBP against the four anthropometric indicators with LOWESS (Locally Weighted Scatterplot Smoothing) lines [18], and linear patterns were observed (Appendix A). For analytic purpose, we computed quartiles of WC, WHtR and CI, and treated them as categorical variables with the first quartile as the reference category. The normal range (18.5–24 kg/m^2^) served as reference category for BMI [17]. Four distinct logistic regression models were fitted to explore the association of the four indicators with HTN. The adjusted models accounted for age, sex, socio-economic status, education, and history of smoking and diabetes simultaneously. Crude and adjusted and odds ratios (ORs) with corresponding 95% confidence intervals (CIs) are reported. Missing data were handled by complete case analysis.

ROC curves were constructed, and area under the curves (AUCs) with 95% CIs retrieved, to evaluate the capability of the four indicators in predicting HTN. An AUC of 1 reflects a perfect predictive capability, whereas AUC ≤ 0.5 suggests the discriminatory power is no better than chance. The ROC curve analysis was sex-specific, and survey weights were not applied. We used the “rocgold” command in Stata that allows estimation of ROC curves and compares each indicator’s ROC curve to that of BMI. The command implements the nonparametric approach proposed by DeLong et al. [19]. Optimal cut-offs were defined as the values of the indicators that maximized the Youden index (J = sensitivity + specificity − 1) [20]. All statistical tests were two-sided, and differences were considered statistically significant at *p* < 0.05.

### 2.6. Ethics Statement

This study was based on secondary analysis of publicly available data from the 2017-18 ADHS. Therefore, no separate ethical approval was sought [15]. The ADHS was approved by the Institutional Review Board at ICF on 11 March 2015 (ICF Project Number: 132989.0.000). The research was carried out in accordance with the Declaration of Helsinki (version 2013).

## 3. Results

Table 1 presents the socio-demographic and anthropometric characteristics of the study participants according to HTN status. Data on BP were available for 20635 participants aged 15–59 years. Mean age of the participants was 38.2 years (standard deviation (SD) 13.5). The prevalence of HTN was 28.6% (28.0–29.2). HTN prevalence was similar across categories of sex and self-reported smoking and alcohol consumption. The prevalence increased with age (*p* for trend < 0.001) and increasing BMI, WC, WHtR and CI (*p* for trend for all four indicators < 0.001). Greater than 70% of those who reported having diabetes were hypertensive as well.

Table 2 shows the overall and sex-specific associations of the four anthropometric indicators with HTN. Compared to participants with normal BMI, the odds of HTN were nearly 1.5 times higher among overweight participants (adjusted odds ratio (aOR): 1.48; 95% CI: 1.30–1.68), and nearly 2.4 times higher among obese participants (aOR: 2.37; 95% CI: 2.05–2.74). WC, WHtR and CI were also positively associated with HTN, as participants in the fourth quartile (Q4) had significantly higher odds compared to participants in the first quartile. The magnitude of association with HTN was relatively lower for CI compared to the other three indicators (aOR for Q4: 1.62; 95% CI: 1.38–1.89). 

Sex modified the association between BMI and HTN among obese participants (*p* for interaction term obese×female < 0.001). Obese females had approximately 2.7 times higher odds of HTN than females with normal BMI (aOR: 2.72; 95% CI: 2.32–3.19). Sex also modified the associations of WC, WHtR and CI with HTN among participants in the respective fourth quartiles (*p* for interaction terms WC Q4 × female, WHtR Q4 × female and CI Q4 × female: 0.001, <0.001 and 0.002, respectively). Females in the fourth quartile of WC, WHtR and CI had approximately 2.9, 2.5- and 1.8-times higher odds of HTN, respectively (Table 2). Appendix A shows the predictive margins of HTN probability for each category of the four indicators, and corroborates the effect measure modifications observed in the highest categories.

Table 3 demonstrates the predictive performance of the four anthropometric indicators based on ROC curve analysis. In females, the highest AUC was recorded for BMI (0.729; 95% CI: 0.720–0.738), followed by WHtR (0.725; 95% CI: 0.716–0.734). The difference between these two AUCs was not statistically significant (*p* = 0.279). The AUC for BMI was significantly higher than that for WC (*p* < 0.001) or CI (*p* < 0.001). The optimal BMI and WHtR cut-offs for identifying females with HTN were 27.01 kg/m^2^ (sensitivity 66.1% and specificity 68.2%) and 0.53 (sensitivity 74.0% and specificity 59.9%), respectively. No statistically significant difference was observed between the AUCs for BMI (0.648; 95% CI: 0.633–0.663) and WHtR (0.637; 95% CI: 0.622–0.652) among males (*p* = 0.227). 

The optimal BMI and WHtR cut-offs for identifying males with HTN were 25.64 kg/m^2^ (sensitivity 68.4% and specificity 53.9%) and 0.54 (sensitivity 52.2% and specificity 68.8%), respectively. Figure 1 below illustrates the sex-specific ROC curves for the four anthropometric indicators.

## 4. Discussion

In a nationally representative sample of Albanians aged 15–59 years, more than one in four had HTN. The anthropometric indicators showed positive associations with HTN. The strength of the associations was substantial for participants in the highest categories of the indicators. Among participants with the highest levels of adiposity, the strength of the associations varied in a sex-specific manner with the odds of HTN being higher for females. BMI and WHtR demonstrated similar discriminatory power, regardless of sex. However, BMI outperformed WC and CI in predicting HTN in both sexes. We also provided optimal cut-offs for the four indicators to identify individuals at risk of HTN.

The observed positive associations of anthropometric indicators with HTN agree with previous studies from Greece [21], Italy [22], Iran [23] and elsewhere [24,25]. The diverse mechanisms linking obesity or increased adiposity with HTN are centered on adipose tissue dysfunction characterized by an overproduction of proinflammatory adipokines (such as leptin), and suppression of anti-inflammatory adipokines (such as adiponectin). This is accompanied by chronic inflammation, oxidative stress, endothelial dysfunction, and untoward activation of renin-angiotensin-aldosterone systems and the sympathetic nervous system. These alterations culminate in development of HTN [26,27].

Sex modified the relationship between the four anthropometric indicators and HTN, whereby females in the highest categories of these indicators had higher likelihood of HTN than males in the same categories. Some previous studies documented similar findings, but mostly for BMI. The association of BMI with HTN was stronger among females in a nationally representative sample (*n* = 11,247) of Australian adults [28]. BMI showed a stronger association with HTN among females in a cross-sectional analysis from Japan (*n* = 4557, age range 35–59) [29]. On the contrary, Chen et al. found a stronger association of BMI with HTN among males in China (*n* = 486,936, age range 30–79) [30]. We could not explore the mechanism(s) underlying this effect measure modification. We posit it could be an indication of hormonally driven differences observed at higher levels of adiposity. The mean age of the females with BMI ≥ 30 kg/m^2^ was 46.9 years (SD 9.9), and those in the fourth quartiles of WC, WHtR and CI were 47.1 years (SD 10.1), 46.9 years (SD 10.1) and 45.1 years (SD 11.8), respectively. Besides, the mean age at natural menopause in Albania is 49.0 (SD 4.9) [31]. While the ADHS lacked data on menopause, a considerable proportion of the females in the highest categories of the four indicators were likely to be perimenopausal [32]. This period is marked by a shift in the hormonal profile as estradiol level decreases and (bioavailable) testosterone increases, leading to higher testosterone-to-estradiol ratio [33]. Age-corrected estradiol levels appear lower also among obese females of reproductive age than their normal-weight peers [34]. Elevated testosterone-to-estradiol ratio is associated with a redistribution of body fat with enhanced central adiposity from accumulation of visceral adipose tissue (VAT) independent of age, ethnicity, physical activity and total body fat [35,36]. As a preferential source of proinflammatory adipokines [35], excessive VAT drives the development of HTN [37,38]. Of note is that this perimenopausal accumulation of VAT coincides with loss of lean mass, resulting in little to no gain in body weight. Consequently, this should be better captured by indicators of central than general obesity [39] such as WC. While WC demonstrated the highest strength of association with HTN (adjusted OR of 2.94 among females in quartile 4), BMI performed better than WC in predicting HTN among females. This may relate to analytic differences because of the quartiles of WC used in logistic regression. The females in the fourth quartile of WC had mean BMI of 33.1 kg/m^2^ (SD 5.0), well above the 27.0 kg/m^2^ optimal cut-off identified in ROC curve analysis. Nonetheless, WC is insensitive to the variation in HTN risk by height [40], as it does not take height into account, and WC was outperformed by BMI in our analysis.

In this study, BMI performed as good as WHtR in predicting HTN in both sexes. This agrees with a pooled analysis of cross-sectional data from 16 Asian cohorts that were part of the Decoda Study (*n* = 20,827) [41]. However, the comparable performance of BMI and WHtR in our analysis contrasts with two meta-analyses showing greater discriminatory power of WHtR compared to BMI: one from 2012 (*n* = 31) [9] and the other from 2018 (*n* = 38) [42]. The pooled AUC for WHtR was 0.732 (95% CI: 0.707–0.757) among females and 0.690 (0.668–0.713) among males in the former, and 0.679 (0.673–0.686) among females and 0.649 (0.641–0.657) among males in the latter. The pooled AUC for BMI was 0.693 (0.659–0.726) among females and 0.654 (0.627–0.682) among males in the former, and 0.656 (0.649–0.662) among females and 0.637 (0.629–0.645) among males in the latter. The 2012 meta-analysis tested the difference in pooled AUCs for WHtR and BMI with the Q statistic, and a significant difference (*p* < 0.05) was observed only among males [9]. Nevertheless, southern European populations were under-represented in these meta-analyses, as one study from Italy and one from Turkey were included, and neither meta-analysis examined the performance of CI. CI performed well in predicting HTN in some populations [11,12], but fared poorly in our analysis, registering the lowest AUCs in both sexes. This conforms with earlier cross-sectional analyses [43,44] where performance of CI in terms of AUC appeared inferior to BMI, WHtR and WC. In line with existing literature [9,42], the AUCs for all indicators appeared higher among females than males, implying that the indicators predicted HTN more precisely among females.

Although many studies have explored the predictive performance of common anthropometric indicators, findings remain varied. De Oliveira and colleagues found no significant differences in the AUCs for BMI and WC in predicting HTN in a Brazilian cohort (*n* = 1627, age range 18–102 years). While their mean BMI was similar to our study, HTN prevalence (40%) and mean WC (87.9 cm among non-hypertensives and 96.2 cm among hypertensives) were higher. The authors did not evaluate the discriminatory powers of WHtR and CI [45]. The AUCs of BMI, WHtR and WC did not differ significantly in earlier studies from Nigeria (*n* = 912, HTN prevalence 22.8%) [44], southern Brazil (*n* = 1720, HTN prevalence among females and males 30.5% and 51.6%, respectively) [43], and in a sample of Filipino-American females aged 40–65 years (*n* = 382, HTN prevalence 50%) [46]. Conversely, WHtR performed better than WC and BMI in predicting prevalent HTN in both sexes in a nationally representative sample from Jordan, with HTN prevalence of 21.4% among females and 28.2% among males [47]. WC yielded significantly higher AUC for HTN than that of BMI in both sexes in a recent cross-sectional study (*n* = 1488) from China in which the prevalence of HTN was 52.6% [37]. The heterogeneity of these findings could be reflective of ethnic differences in the regional distribution of body fat [48,49], differences in the capability of different indicators in capturing obesity between populations, or differences in statistical methods applied.

The optimal cut-offs for BMI identified in this study (27.01 among females and 25.64 kg/m^2^ among males) were lower than the 30 kg/m^2^ WHO cut-off for defining obesity [17], and those reported for Peruvian [12] and Jordanian adults [47]; but higher than those reported in studies from China [37,50], South Korea [51] and Nigeria [44]. The optimal WC cut-off among males in this analysis (86.25 cm) was lower than the 94 cm International Diabetes Federation (IDF) cut-off defining central adiposity. By contrast, the WC cut-off for females (91.05 cm) was higher than the 80 cm IDF cut-off for females [52]. The WHtR cut-offs were similar in both sexes (0.53 among females and 0.54 among males) and supported the proposition of considering 0.5 a universal cut-off [9]. 

Our findings delineated the burden of obesity-related HTN, often recognized as a distinct hypertensive phenotype [27]. This has serious public health implications for Albania. The prevalence of HTN in our representative sample neared 30%, and that of BMI-defined overweight and obesity among hypertensives exceeded 80% (Table 1). A meta-analysis of 239 prospective studies demonstrated that the lowest risk of all-cause mortality is associated with a BMI level of 20–25 kg/m^2^, and above that mortality increases almost log-linearly with BMI [53]. On average, the hypertensives had a BMI of 28.84 kg/m^2^ and the normotensives had 25.50 kg/m^2^ in this study, suggesting a population level shift that could drive obesity-related HTN and associated mortality in the transitioning, upper-middle-income context of Albania [54]. However, HTN awareness and control are suboptimal in Albania [14]. Based on our findings, anthropometric indicators, particularly BMI and WHtR, could be promising in identifying Albanians with heightened risk of HTN. Apart from CI, the AUCs surpassed 0.70 among females and 0.60 among males. Despite these modest AUCs, BMI and WHtR can be used satisfactorily as initial, population-based screening measures [9], given the reasonable sensitivity estimates of the identified cut-offs. For instance, the sensitivity exceeded 65% in both sexes for the respective BMI cut-offs (Table 3). The individuals identified using the cut-offs can, thereafter, be offered structured screening as recommended in the 2018 ESC/ESH Guideline [16]. Owing to simplicity and extensive use in practice, BMI and WHtR have the potential to be integrated in primary care across the country. Furthermore, the electronic health record system administered in Albania in 2016 [55] can be leveraged to pool information on these indicators. The data-driven cut-offs from this study could inform the formulation of a nationwide strategy to identify individuals at risk. Given the surge in obesity-related noncommunicable diseases in middle-income countries [1], policymakers in Albania need to prioritize strategies contingent on simple, cost-effective measures to circumvent the challenges of population-wide control of HTN [56]. 

Certain limitations and strengths of the present study need to be considered for critical appraisal of the findings. The study drew on a cross-sectional analysis of secondary data, and the anthropometric indicators are surrogate measures of adiposity. Therefore, causal inferences cannot be drawn, and prospective studies using direct measures of adiposity are warranted for pinpointing the predictive performance of the indicators among Albanians. The relationship of the cut-offs identified in this study with such endpoints as cardio-vascular and all-cause mortality in Albania needs to be investigated in future. Moreover, the meta-analysis by Deng et al. showed that study design may affect the AUCs registered by anthropometric indicators [42]. Data on physical activity were not collected in the ADHS and its potential role as a confounder [57] could not be controlled. Residual confounding cannot be entirely ruled out too. Nonetheless, core strengths of this study include a large sample representative at national and subnational levels, identification of actionable cut-offs for common anthropometric indicators and highly standardized procedures for anthropometric data collection adopted by The DHS Program.

## 5. Conclusions

In conclusion, our findings suggested substantial, positive associations of select anthropometric indicators of general and abdominal obesity (BMI, WHtR, WC and CI) with HTN. Sex modified the relationship of the indicators with HTN, whereby the associations were significantly stronger among females than males, but only in the highest categories of all four indicators. Regardless of sex, there was no significant difference in the AUCs yielded by BMI and WHtR, implying their comparable performance in predicting HTN among Albanians. BMI outperformed WC and CI in terms of AUC. Future studies of prospective design with direct assessment of adiposity could help better quantify the obesity-HTN relationship and the discriminatory power of the indicators. The optimal cut-offs for the anthropometric indicators identified in this study could form the basis for a public health initiative to identify individuals at heightened risk of HTN, and strengthen HTN control among Albanians aged 15–59 years.

## Figures and Tables

**Figure 1 nutrients-13-03373-f001:**
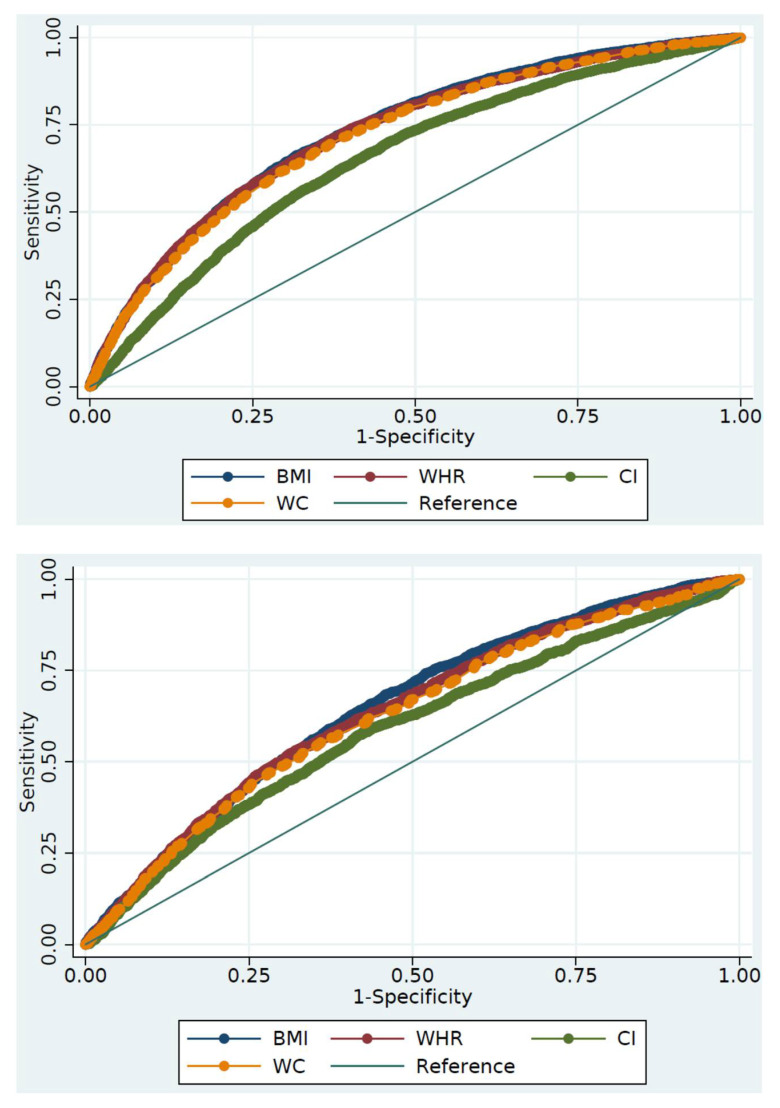
Sex-specific receiver-operating characteristic (ROC) curves for the four anthropometric indicators. Sensitivity entails the true positives and 1-specificity indicates the false positives. The first panel (**top**) shows the curves for females, and the second panel (**bottom**) shows the curves for males.

**Table 1 nutrients-13-03373-t001:** Sample characteristics by hypertension status.

Characteristics	Unweighted Frequency	Weighted Frequency	Non-Hypertensive Weighted % (95% CI) or Weighted Mean (SD)	Hypertensive Weighted % (95% CI) or Weighted Mean (SD)
Overall	20,635	19,591	71.4 (70.8–72.0)	28.6 (28.0–29.2)
Age (years)				
15–19	2354	2253	93.4 (92.4–94.5)	6.6 (5.5–7.6)
20–29	4187	4318	89.3 (88.3–90.2)	10.7 (9.8–11.7)
30–39	3882	3645	82.8 (81.6–84.1)	17.2 (15.9–18.4)
40–49	4587	4215	64.7 (63.3–66.2)	35.3 (33.8–36.7)
50–59	5625	5160	44.1 (42.8–45.5)	55.9 (54.5–57.2)
Sex				
Female	14,718	10,248	71.7 (70.8–72.6)	28.3 (27.4–29.2)
Male	5917	9343	71.1 (70.1–72.0)	29.0 (28.0–29.9)
Household wealth				
Poorest	6794	4753	67.7 (66.3–69.0)	32.3 (31.0–33.7)
Middle-status	6879	6029	69.2 (68.0–70.3)	30.8 (29.7–32.0)
Richest	6962	8809	74.9 (74.0–75.8)	25.1 (24.2–26.0)
Educational status				
Primary or below	9675	7953	65.5 (64.5–66.6)	34.5 (33.4–35.5)
Secondary	7741	7925	71.2 (70.2–72.2)	28.8 (27.8–29.8)
Above secondary	3210	3701	84.4 (83.2–85.6)	15.6 (14.4–16.8)
Residence				
Urban	9479	11,249	73.2 (72.3–74.0)	26.8 (26.0–27.7)
Rural	11,156	8342	69.0 (68.0–70.0)	31.0 (30.0–32.0)
Self-reported Diabetes				
Yes	324	312	27.9 (22.9–32.9)	72.1 (67.1–77.1)
No	20,311	19,279	72.1 (71.5–72.7)	27.9 (27.3–28.5)
H/O smoking				
Smoker	2479	3786	69.4 (67.9–70.9)	30.6 (29.1–32.1)
Nonsmoker	18,156	15,805	71.9 (71.2–72.6)	28.1 (27.4–28.8)
H/O alcohol consumption				
Yes	6559	8483	70.8 (69.8–71.7)	29.2 (28.3–30.2)
No	14,076	11,108	71.8 (71.0–72.7)	28.2 (27.3–29.0)
Body mass index (BMI, kg/m^2^)	20,231	18,950	25.50 (4.58)	28.84 (5.19)
BMI categories				
Underweight (<18.5)	499	473	92.4 (90.0–94.8)	7.6 (5.2–10.0)
Normal (18.5–24.9)	8164	7606	84.0 (83.2–84.8)	16.0 (15.2–16.8)
Overweight (25.0–29.9)	7087	6848	67.9 (66.8–69.0)	32.1 (31.0–33.2)
Obese (≥30)	4481	4023	50.5 (49.0–52.1)	49.5 (47.9–51.0)
Waist circumference (WC, cm)	20,072	18,693	85.00 (13.99)	93.96 (13.46)
WC (cm) quartiles				
Quartile 1 (< 75)	4454	3626	89.2 (88.1–90.2)	10.8 (9.8–11.9)
Quartile 2 (75–85)	4972	4380	80.6 (79.4–81.8)	19.4 (18.2–20.6)
Quartile 3 (85–96)	5588	5544	68.1 (66.9–69.4)	31.9 (30.6–33.1)
Quartile 4 (>96)	5058	5143	53.4 (52.1–54.8)	46.5 (45.2–47.9)
Waist-to-height ratio (WHtR)	20,039	18,650	0.51 (0.09)	0.57 (0.09)
WHtR quartiles				
Quartile 1 (<0.46)	4765	4317	88.2 (87.2–89.1)	11.8 (10.9–12.8)
Quartile 2 (0.46–0.52)	5085	4872	78.9 (77.7–80.0)	21.1 (20.0–22.3)
Quartile 3 (0.52–0.58)	4596	4420	67.6 (66.2–68.9)	32.4 (31.1–33.8)
Quartile 4 (>0.58)	5593	5041	52.1 (50.7–53.5)	47.9 (46.5–49.3)
Conicity index (CI)	20,000	18,633	1.20 (0.13)	1.25 (0.12)
CI quartiles				
Quartile 1 (<1.12)	4933	4168	83.9 (82.9–85.1)	16.1 (14.9–17.2)
Quartile 2 (1.12–1.20)	4818	4277	77.3 (76.0–78.6)	22.7 (21.4–24.0)
Quartile 3 (1.20–1.29)	5341	5202	67.6 (66.4–68.9)	32.4 (31.1–33.6)
Quartile 4 (>1.29)	4908	4986	58.7 (57.4–60.1)	41.3 (39.9–42.6)

CI: confidence interval, SD: standard deviation, H/O: history of. Missing data on education (*n* = 9), BMI (*n* = 404), WC (*n* = 563), WHtR (*n* = 596), CI (*n* = 635).

**Table 2 nutrients-13-03373-t002:** Logistic regression analyses of association of the four anthropometric indicators with hypertension.

Variables	All		Male		Female	
	Crude OR (95% CI)	Adjusted ^1^ OR (95% CI)	Crude OR (95% CI)	Adjusted ^2^ OR (95% CI)	Crude OR (95% CI)	Adjusted ^2^ OR (95% CI)
BMI (kg/m^2^) categories						
Underweight (<18.5)	0.43(0.26–0.70) *	0.87 (0.51–1.48)	0.57 (0.23–1.39)	1.03 (0.38–2.76)	0.38 (0.20–0.65)	0.91 (0.49–1.71)
Normal (18.5–24.9)	Ref.	Ref.	Ref.	Ref.	Ref.	Ref.
Overweight (25.0–29.9)	2.59 (2.33–2.87) *	1.48 (1.30–1.68) *	2.21 (1.85–2.64) *	1.50 (1.24–1.81) *	2.73 (2.41–3.09) *	1.36 (1.17–1.56) *
Obese (≥30)	5.92 (5.23–6.70) *	2.37 (2.05–2.74) *	3.33 (2.65–4.19) *	1.83 (1.45–2.32) *	7.18 (6.25–8.25) *	2.72 (2.32–3.19) *
Waist circumference (cm) quartiles						
Q1 (<75)	Ref.	Ref.	Ref.	Ref.	Ref.	Ref.
Q2 (75–85)	1.98 (1.69–2.32) *	1.32 (1.10–1.59) *	1.42 (1.03–1.94) *	1.08 (0.78–1.50)	2.34 (1.96–2.80) *	1.30 (1.03–1.64) *
Q3 (85–96)	3.84 (3.30–4.47) *	1.90 (1.56–2.30) *	2.61 (1.92–3.56) *	1.54 (1.09–2.18) *	4.89 (4.14–5.76) *	1.74 (1.40–2.16) *
Q4 (>96)	7.15 (6.07–8.41) *	2.69 (2.22–3.26) *	4.25 (3.12–5.78) *	1.95 (1.38–2.75) *	10.88 (9.19–12.88) *	2.94 (2.36–3.65) *
Waist-to-height ratio categories						
Q1 (<0.46)	Ref.	Ref.	Ref.	Ref.	Ref.	Ref.
Q2 (0.46–0.52)	2.00 (1.70–2.36) *	1.30 (1.06–1.59) *	1.86 (1.44–2.41) *	1.38 (1.03–1.84) *	2.03 (1.66–2.47) *	1.08 (0.84–1.38)
Q3 (0.52–0.58)	3.59 (3.07–4.20) *	1.63 (1.33–2.01) *	3.17 (2.47–4.07) *	1.70 (1.25–2.32) *	3.98 (3.31–4.78) *	1.42 (1.10–1.80) *
Q4 (>0.58)	6.87 (5.87–8.03) *	2.36 (1.95–2.86) *	4.32 (3.35–5.56) *	1.91 (1.42–2.57) *	10.07 (8.46–11.98) *	2.46 (1.96–3.07) *
Conicity index quartiles						
Q1 (<1.12)	Ref.	Ref.	Ref.	Ref.	Ref.	Ref.
Q2 (1.12–1.20)	1.54 (1.33–1.78) *	1.20 (1.02–1.41) *	1.20 (0.92–1.56)	1.05 (0.80–1.46)	1.83 (1.56–2.18) *	1.24 (1.02–1.50) *
Q3 (1.20–1.29)	2.50 (2.17–2.90) *	1.43 (1.21–1.69) *	1.79 (1.40–2.30) *	1.25 (0.95–1.66)	3.24 (2.76–3.81) *	1.44 (1.18–1.76) *
Q4 (>1.29)	3.67 (3.19–4.23) *	1.62 (1.38–1.89) *	2.47 (1.93–3.16) *	1.34 (1.02–1.74) *	5.25 (4.30–6.13) *	1.79 (1.48–2.13) *

OR: odds ratio, CI: confidence interval, Q: quartile, Ref.: reference category. * indicates statistical significance as CI excludes 1. ^1^ Adjusted for age, sex, socio-economic status, education, history of smoking, and self-reported diabetes. ^2^ Adjusted for age, socio-economic status, education, history of smoking, and self-reported diabetes.

**Table 3 nutrients-13-03373-t003:** Performance of the four anthropometric indicators as predictors of hypertension from ROC curve analysis in the unweighted sample.

Indicators	AUC (95% CI)	*p*-Value ^1^	Youden’s Index	Optimal Cut-Off	Sensitivity (%)	Specificity (%)
Females						
BMI (kg/m^2^)	**0.729 (0.720–0.738)**	Ref.	**0.343**	27.01	66.1	68.2
WC (cm)	0.718 (0.709–0.727)	<0.001	0.327	91.05	67.1	65.6
WHtR	0.725 (0.716–0.734)	0.279	0.338	0.53	74.0	59.9
CI	0.653 (0.643–0.663)	<0.001	0.242	1.24	70.1	54.0
Males						
BMI (kg/m^2^)	**0.648 (0.633–0.663)**	Ref.	**0.223**	25.64	68.4	53.9
WC (cm)	0.626 (0.611–0.642)	0.002	0.192	86.25	55.3	64.0
WHtR	0.637 (0.622–0.652)	0.227	0.209	0.54	52.2	68.8
CI	0.589 (0.573–0.605)	<0.001	0.156	1.19	57.9	57.7

BMI: body mass index, WC: waist circumference, WHtR: waist-to-height ratio, CI: conicity index, ROC: receiver operating characteristic, AUC: area under the ROC curve. The highest values of AUC and Youden’s index in bold. ^1^ Bonferroni-adjusted *p*-value for comparison with AUC for BMI.

## Data Availability

Datasets are available upon request from The DHS Program website: https://www.dhsprogram.com/data/dataset/Albania_Standard-DHS_2017.cfm?flag=0. The datasets analyzed during the current study are available on reasonable request.

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
