# Peer review of "Relationship of Anthropometric Indicators of General and Abdominal Obesity with Hypertension and Their Predictive Performance among Albanians: A Nationwide Cross-Sectional Study"

_nutrients, 2021, doi:10.3390/nu13103373_

Round 1

Reviewer 1 Report

They appear in the attached file.

Reviewer 2 Report

 The current paper under the title “Relationship of Anthropometric Indicators of General and Abdominal Obesity with Hypertension and Their Predictive Performance among Albanians: A Nationwide Cross-Sectional study” is presented by Mohammad Redwanul Islam, Md. Moinuddin, Samaha Masroor Saqib and Syed Moshfiqur Rahman from the  Department of Women´s and Children´s Health, Uppsala University in Uppsala (Sweden), the  Faculty of Health, Social Care & Medicine, Edge Hill University,  in Ormskirk (UK) and the Institute of Child Health, University College London in London (UK).

 In this cross-sectional study including a sample of 20,635 Albanians aged 15–59 years, the authors aimed to “to explore the relationship of BMI, WHtR, WC and CI with HTN among Albanians, to examine the performance of these indicators in predicting HTN from analysis of receiver-operating characteristic (ROC) curves, and to determine their optimal cut-offs for predicting the likelihood of HTN”. Main conclusion obtained was that there is a “positive associations of select anthropometric indicators of general and abdominal obesity (BMI, WHtR, WC and CI) with HTN.”

 Overall, the results here-in reported suggest that “surrogate markers of abdominal adiposity, mainly WHtR, are good markers to identify individuals at high risk for developing HTN”.

Comments

This is a nice study which has been carried out rigorously. The study provides evidence of the usefulness of general and abdominal anthropometric markers as predictors of arterial hypertension in Albanian population. The draft is well presented, and the English style is good, except for few minor errors such as:

Lines 79-81: the sentence is somehow confusing, please rewrite or at least add a comma between those and underwent”.

Line 231: “…overproduction production of proinflammatory…”. Please delete “production”.

Line 290: “…findings remained varied. De Oliveira and colleagues…”

 The data analysis has been performed correctly and results are well displayed through the draft. Authors present logistic regressions and ROC curves which are crucial to answer the objectives. They have also presented nice supplementary figures that support the conclusions.

Otherwise, limitations have been acknowledged by authors and the discussion is undoubtedly interesting and well written, although it is too long and could be summarized.

 Regardless the nice study carried out, I have one concern. Authors have performed the analysis with conicity index, which as discussed by authors in the Introduction section, is lacking established cut-off limits and have not enough evidence. By contrast, the body shape index (ABSI) that normalizes WC to be uncorrelated with BMI, is a predictor of cardiovascular risk (CVR) in a variety of studies in different populations, showing correlation with total as well as CV mortality (Krakauer NY, Krakauer JC. The new anthropometrics and abdominal obesity: A body shape index, hip index, and anthropometric risk index. In: Watson RR (ed). Nutrition in the Prevention and Treatment of Abdominal Obesity, 2nd ed, Elsevier; 2019: 19–27.) Testing this anthropometric index as a possible predictor of hypertension, would add value to this work.

Minor issues.

 In my opinion, due to the large number of individuals included in this study, I believe that it is unlikely that the fact that BP measurement and anthropometric indicators were only assessed in only 50 % of men, would change the results. Nevertheless, authors should explain what the rationale for this procedure was.

 Urban and rural areas should be defined. Usually, rural areas are those with less than 5,000 subjects.  Former smokers (quit smoking > 1 year) category should be included or at least included in the analysis together with non-smoker category, while those (< 1 year after quitting smoking) could be considered current smokers.  

 Lines 141 to 143: “The adjusted models accounted for age, sex, socio-economic status, education, and history of diabetes simultaneously.” History of smoking is missing in this sentence.
